# Estimating Suspended Sediment Concentration Using Remote Sensing for the Teles Pires River, Brazil

**Rhavel Salviano Dias Paulista** [1], **Frederico Terra de Almeida** [2,*], **Adilson Pacheco de Souza** [2],
**Aaron Kinyu Hoshide** [3,4], **Daniel Carneiro de Abreu** [2,3], **Jaime Wendeley da Silva Araujo** [2]
**and Charles Campoe Martim** [5]

1   Environmental Sciences, Federal University of Mato Grosso, Sinop 78557-287, MT, Brazil
2   Institute of Agrarian and Environmental Sciences, Federal University of Mato Grosso,
    Sinop 78557-287, MT, Brazil
3   AgriSciences, Institute of Agrarian and Environmental Sciences, Federal University of Mato Grosso,
    Avenida Alexandre Ferronato, 1200, Sinop 78555-267, MT, Brazil
4   College of Natural Sciences, Forestry and Agriculture, University of Maine, Orono, ME 04469, USA
5   Postgraduate Program in Environmental Physics, Federal University of Mato Grosso,
    Cuiabá 78060-900, MT, Brazil
*   Correspondence: fredterr@gmail.com or frederico.almeida@ufmt.br; Tel.: +55-66-99995-1315

**Abstract:** Improving environmental sustainability involves measuring indices that show responses to different production processes and management types. Suspended sediment concentration (SSC) in water bodies is a parameter of great importance, as it is related to watercourse morphology, land use and occupation in river basins, and sediment transport and accumulation. Although already established, the methods used for acquiring such data in the field are costly. This hinders extrapolations along water bodies and reservoirs. Remote sensing is a feasible alternative to remedy these obstacles, as changes in suspended sediment concentrations are detectable by satellite images. Therefore, satellite image reflectance can be used to estimate SSC spatially and temporally. We used Sentinel-2 A and B imagery to estimate SSC for the Teles Pires River in Brazil's Amazon. Sensor images used were matched to the same days as field sampling. Google Earth Engine (GEE), a tool that allows agility and flexibility, was used for data processing. Access to several data sources and processing robustness show that GEE can accurately estimate water quality parameters via remote sensing. The best SSC estimator was the reflectance of the B4 band corresponding to the red range of the visible spectrum, with the exponential model showing the best fit and accuracy.

**Keywords:** Amazonia; Google Earth Engine; hydro-sedimentology; reflectance; satellite imagery

## 1. Introduction

The search for environmentally sustainable production processes requires establishing parameters to evaluate production systems such as those related to forestry and agriculture. These factors can enable development while mitigating adverse environmental impacts. Among them, suspended sediment concentration in water bodies is important for evaluating such land use and its impacts on water resources. Suspended sediment concentration (SSC) is relevant to assess the quality of water bodies, as it is directly connected to the morphology of channels and silting processes in reservoirs [1,2]. Although erosion and particle sedimentation are natural phenomena, anthropogenic actions can enhance these processes, causing loss of water quality, silting up of water bodies, and reducing the useful life of reservoirs [3,4]. Although traditional methods for acquiring sedimentological data are reliable, alternative techniques are needed to speed up and improve SSC quantification, making it less costly [5]. Another problem of traditional methods is the difficulty in establishing continuous observations, impairing long-term SSC assessments. Furthermore, the

site-specific nature of sampling makes these methods even more difficult to extrapolate values to other locations along the water body [4].

Remote sensing is an alternative for SSC determination. At first, its application was limited to oceans or large water bodies, but as higher spatial-resolution orbital sensors emerged, remote sensing technology could be used for smaller bodies of water [6]. As SSC increases, the surface reflectance of water bodies also rises, enabling the detection of its variations along a water body [7]. One advantage of sedimentological monitoring using satellite images is the potential to monitor sediment inflow sites in reservoirs, which are hard to monitor [3]. Therefore, satellite imagery can be used to monitor several points of water bodies simultaneously, including hard-to-reach areas. Satellites can provide temporal resolution at a daily time scale which can vary depending on the type of sensor used. Another benefit of remote sensing is the ease of accessing data from services such as the U.S. Geological Survey (USGS)-Earth Explorer and the Copernicus Science Hub. These platforms provide free satellite images to the scientific community. Another important platform is Google Earth Engine (GEE) which, in addition to making data available, allows data processing which has improved access to high-performance computing [8,9].

There are three main remote sensing approaches for water quality characterization, namely: empirical, semi-empirical, and analytical approaches [10]. In an empirical approach, simple or multiple regressions are performed between reflectance values from satellite images and water quality parameters. In a semi-empirical one, however, the spectral behavior of parameters studied must be measured, in loco or in the laboratory, with specific spectral bands being selected to capture such response. Finally, an analytical approach is a physical evaluation in which specific inherent optical properties are studied throughout the entire water column, and, subsequently, these characteristics are related to the apparent optical properties of water quality parameters.

Several authors have used satellite images to quantify suspended sediment concentration (SSC) [11–16]. Using the empirical approach, previous research found strong relationships between SSC and the Normalized Difference Water Index (NDWI) radiometric index for the Teles Pires River [17]. Another study applied this method to two different orbital platforms, Landsat and Sentinel-2 A and B, and evaluated several spectral bands and radiometric indices, finding good SSC estimators for the Doce River in southeast Brazil [18]. Both authors faced the problem of synchronizing the dates of satellite images with those for field sampling. Therefore, they opted to insert images from days after and before sampling, even though there may have been variations in flow rates. Two studies have successfully applied the semi-empirical method with good results relating to responses obtained by spectrometers in the field with images from the Multi-Spectral Instrument (MSI) and Moderate Resolution Imaging Spectroradiometer (MODIS) sensors [19,20]. Another study applied this analytical process to understand the behavior of the optical properties of water along the Amazon basin [4]. Using the same procedure, other researchers built models to estimate suspended matter in Lake Frisian in the northeastern part of The Netherlands [21].

Regardless of the approach used, satellite imagery reflectance values need to be referring to water quality parameters and not to noise caused by the atmosphere. The author emphasized that there is a need for atmospheric calibrations, which seek to recover surface reflectance by subtracting the atmospheric contributions from the reflectance at the top of the atmosphere captured by the orbital sensors [6]. With the spread and popularization of remote sensing data, atmospheric calibration algorithms were developed to minimize such noise; among them is the atmospheric corrector Dark Spectrum Fitting [9]. Dark Spectrum Fitting (DFS) was originally developed for images with metric resolution, but satisfactory results have been obtained when implementing DFS in images from Landsat and Sentinel-2 satellites, which have decametric resolution [22]. DFS is contained within ACOLITE, a multi-sensor atmospheric calibration processor developed by the Royal Belgian Institute of Natural Sciences (RBINS) for aquatic satellite imagery applications.

Another relevant factor is sun glint, which is the reflection of direct sunlight towards remote sensors' field-of-view, preventing proper capture of reflectance from water bodies [23]. Sun glint is a recurrent problem in studies of water surfaces via remote sensing, and some authors have been studying and proposing solutions [24–26]. One solution was implemented in an ACOLITE processor using a sun glint extraction method for the short wave infrared region (SWIR) based Multi-Spectral Instrument (MSI) sensor [27]. In another study, sensors with the SWIR band facilitate atmospheric calibration because water reflectance is equal to zero at this wavelength and under ideal conditions [28]. Moreover, the use of the SWIR band as a parameter for sun glint extraction improves the accuracy of remotely sensed data [27]. By synchronizing field data with satellite passes, remote-sensing products can be validated to assess the presence of sediments [10].

The goal of our research was to evaluate the Google Earth Engine (GEE) platform, both in terms of manipulation and image/data processing, through a semi-automatic method. In doing so, we have three objectives. The first objective of our research was to evaluate empirical models using spectral bands and radiometric indices obtained through images from Sentinel-2 A and B sensors. Second, we sought to estimate the suspended sediments concentration (SSC) in the Teles Pires River basin in Brazil. The third objective of our study was to optimize monitoring mechanisms that help in the management of watersheds and the implementation of public policies.

## 2. Materials and Methods

### 2.1. Study Area

Remotely sensed data were used, and field sampling was conducted in the Teles Pires River. The Teles Pires River basin has an area of 141,524 square kilometers and is located between south latitudes 7°16′47″ to 14°55′17″ and west longitudes 53°49′46″ to 58°7′58″. The Teles Pires River basing watershed covers both of the Brazilian states of Mato Grosso and Pará and is within the Amazon Hydrographic Region. Due to its great length, the Teles Pires river basin is commonly divided into upper, medium, and lower regions. In the lower and medium Teles Pires, the predominant biome is the Amazon forest, while the Cerrado (i.e., savannah) is the predominant biome in the upper basin further south. The basin has two climates according to the Köppen's classification. In the upper region and most of the medium basin, it is classified as Aw, which stands for tropical, with dry winters and rainy summers. Meanwhile, the area near the Teles Pires River's mouth, where it merges with the Amazon River, has an Am climate, which is a humid tropical climate with short dry seasons and higher amounts of rainfall [29]. The Teles Pires River currently houses four hydroelectric plants (HEP), namely the São Manuel, Teles Pires, Colíder Pesqueiro do Gil, and the Sinop plants (Figure 1).

The São Manuel hydroelectric plant is located in the lower basin, at coordinates 9°11′11.06″ South and 57°3′1.13″ West and has an installed capacity of 700 megawatts (MW) and a reservoir area of 66 square kilometers (km$^2$). The Teles Pires hydroelectric plant is located at coordinates 9°21′04″ South and 56°46′39″ West and has an installed capacity of 1820 MW and a reservoir area of 150 km$^2$. The Colíder Pesqueiro do Gil plant is located at coordinates 10°59′06.62″ South and 55°45′52.06″ West with an installed capacity of 300 MW and a reservoir area of 114.9 km$^2$. Finally, the Sinop hydroelectric plant (11°16′1″ South, 55°27′14″ West) has an installed capacity of 401.88 MW and a reservoir area of 342 km$^2$ [30].

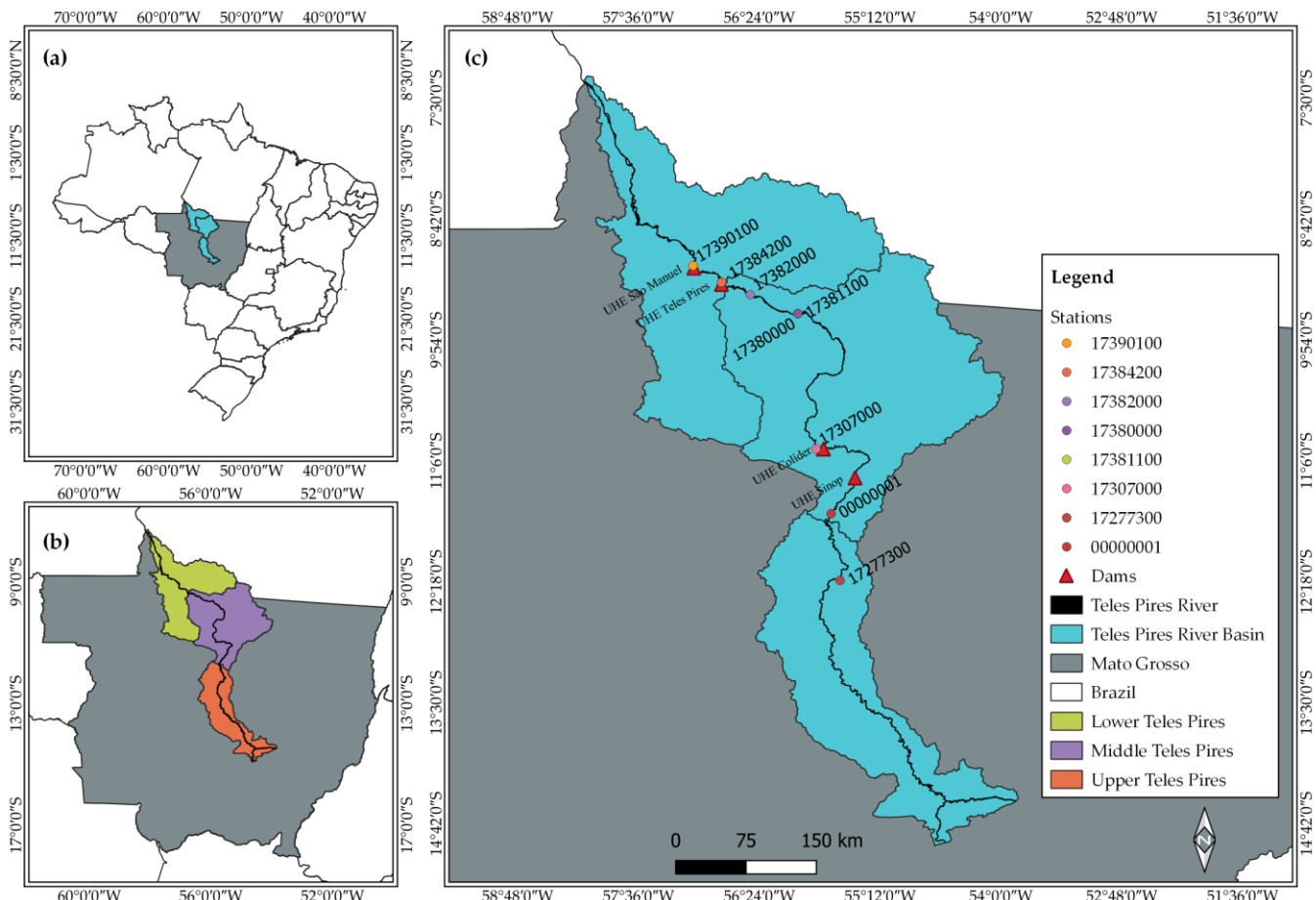

**Figure 1.** Maps showing the study area and dams for hydroelectric stations: (**a**) river basin highlighted (in blue), inserted in the state of Mato Grosso (in grey) in Brazil, (**b**) Teles Pires river basin upper, middle, and lower divisions from north to south and (**c**) location of dams/stations and in-field sampling locations used for field research with respective codes.

*2.2. Suspended Sediment Concentration Data*

Suspended sediment concentration (SSC) data used in this study were obtained from two sources. The first data source is the authors themselves, who performed five data collection campaigns on the same day the Sentinel-2 A and B satellites passed within a section near the Sinop power plant reservoir. The second data source is the National Agency for Water and Basic Sanitation (Agência Nacional das Águas e Saneamento Básico—ANA) [31]. A joint resolution known as ANA/ANEEL n° 03/2010 establishes the monitoring of several hydrological parameters by power utilities that operate the hydroelectric complexes, one of which is a hydro-sedimentological survey. In addition to the operators, the Brazilian Geological Survey (Serviço Geológico do Brasil—CPRM) also monitors this parameter within a section of the Teles Pires River. Under these conditions, all stations in operation between 2016 and 2021 that had solid discharge data were selected. Table 1 shows the information on the section of the Teles Pires River studied by the authors and on the seven stations used in this study that were obtained from ANA that met our requirements.

For solid discharge collection, both authors and HEP operators used the equal width increment (EWI) sampling method with vertically integrated samples [5]. According to Oliveira Carvalho 2008 [5], the EWI represents the average suspended sediment concentration (SSC) in the river section studied. In this method, the section is divided into verticals with the same distance between them. In each vertical, subsamples that integrate the SSC from the surface to the river bottom are collected. All the subsamples are gathered to form a single sample capable of representing the average SSC in the investigated section. This

is commonly called a composite sample. Here, each composite sample is the sum of the water sample volumes acquired in each vertical (Figure 2).

**Table 1.** Characteristics and location of the hydro-sedimentological monitoring stations used in sampling of the Teles Pires River, Mato Grosso state, Brazil.

| Code | Name | Operator | Location | |
|---|---|---|---|---|
| | | | South | West |
| 00000001 | Section Curio | Authors | 11°37′35″ | 55°41′23.37″ |
| 17277300 | HEP Sinop upstream 1 | HEP Sinop | 12°17′17″ | 55°36′03″ |
| 17390100 | HEP São Manuel downstream 1 | HEP São Manuel | 09°09′55″ | 57°03′39″ |
| 17307000 | HEP Colíder Pesqueiro do Gil | HEP Colíder | 10°58′59″ | 55°46′06″ |
| 17380000 | Downstream the mouth of Peixoto de Azevedo | CRRM | 09°38′26″ | 56°01′10″ |
| 17381100 | HEP Teles Pires upstream 2 | HEP Teles Pires | 09°38′23″ | 56°01′09″ |
| 17382000 | HEP Teles Pires upstream 1 | HEP Teles Pires | 09°27′11″ | 56°29′32″ |
| 17384200 | HEP Teles Pires downstream | HEP Teles Pires | 09°19′52″ | 56°46′41″ |

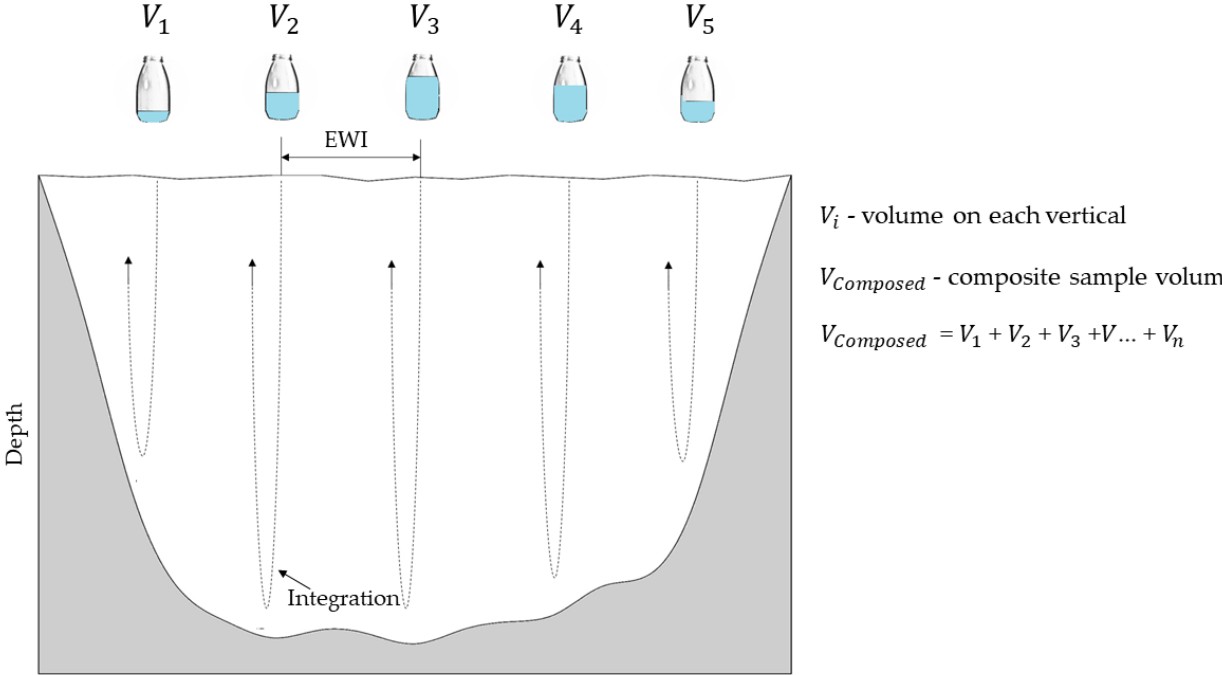

$V_i$ - volume on each vertical

$V_{Composed}$ - composite sample volume

$V_{Composed} = V_1 + V_2 + V_3 + V \ldots + V_n$

**Figure 2.** Description and details of the equal width increment (EWI) method.

With regard to operators, ANA/ANEEL resolution n° 03/2010 requires a quarterly collection frequency, comprising four annual measurements in periods of drought and flood, as well as rising and receding waters. Since stations downstream of the mouth of the Peixoto de Azevedo (17380000) and the second upstream station of the Teles Pires River (17381100) are at a similar location (Table 1), this provided a greater volume of data for each location. The raw data used in this study can be found summarized in Appendix A.

### 2.3. Remote Sensing Data

Remote sensing images were processed in Google Earth Engine (GEE). GEE is a cloud computing platform with a vast orbital data collection allowing for remote sensing data to be processed [8]. GEE is a tool that allows for more open access to high-performance computing and can process large amounts of remote sensing data. We used the Python-based package *geemap* [32] to manipulate and visualize GEE data. One of the collections

available in GEE is called COPERNICUS, which comprises images from Multi-Spectral Instrument (MSI) sensor on board the Sentinel-2 A and B satellite.

This satellite is part of the COPERNICUS earth monitoring program, which is coordinated by the European Commission (EC) and the European Space Agency (ESA). The MSI sensor has a 5-day time resolution and 13 spectral band spans. However, we only used bands B3, B4, and B8, which correspond to green, red, and infrared bands, respectively. Such bands have a 10-m spatial resolution. Band B3 is centered at the 560-nanometer (nm) wavelength and 35-nm width. Band B4 is centered at the 665-nm wavelength and 30-nm width, while band B8 is centered at the 842-nm wavelength and 20-nm width [33]. The main reason for choosing the sensor was its time resolution, which has a 5-day revisit interval. This increases the chances of capturing images on the same day as field data sampling. Bands were chosen based on the reflectance bands that capture the range of suspended sediment concentration [34]. Low suspended sediment concentration (SSC) increases green reflectance, while high SSC increases red and infrared reflectance values.

In addition to testing bands alone, two radiometric indexes that work within these spectral bands were also tested. These were the Normalized Difference Vegetation Index and Normalized Difference Water Index. The Normalized Difference Vegetation Index (NDVI) is the normalized ratio between red and near-infrared bands [35]. We compared the physical behavior of healthy and unhealthy plants within the electromagnetic spectrum to determine vegetation status. The other index, the Normalized Difference Water Index (NDWI), is similar to the NDVI and uses the green instead of red band in order to outline aquatic environment features and highlight water bodies in satellite images [36]. NDWI ranges from $-1$ to 1, with positive values corresponding to aquatic environments.

Satellite images are available in collections, each corresponding to a processing level. We used the second level of processing; that is, products were geometrically corrected and calibrated for the atmosphere to present surface reflectance. However, GEE images were used only for exploratory analysis, identifying cloud-free images with all quality parameters. Using station coordinates and in situ collection dates, an algorithm was developed via the GEE platform, which verified Sentinel-MSI sensor images for the same days of in situ collections. Of the 122 dates, the algorithm verified 31 images corresponding to collection days. From visual inspection, clouds were verified in 16 images, with only 15 images not having cloud obstruction and fitting all quality parameters. The sum of these images combined with those obtained during the in situ campaigns totaled 20 images. Of these 20 images, 14 images were reserved for the creation of the model, with the remaining 6 images used for application of the model. The choice of images for model application took into account the representativeness of the Teles Pires River, selecting at least one image for each studied station. Figure 3 shows the satellite images used for this study, dates, and respective tiles, as well as identifies which images were used to create and validate the models that were developed.

### 2.4. Atmospheric Calibration

Atmospheric calibration seeks to recover surface reflectance by extracting atmospheric contributions from reflectance at the top of the atmosphere [6]. To do so, we used the Dark Spectrum Fitting (DSF) algorithm proposed by a previous study [9]. The DSF is implemented in the ACOLITE processor and was initially developed for metric-resolution satellite images. However, good results can be obtained by applying DSF to decametric-resolution satellite images, where DSF is based on two assumptions for atmospheric contribution estimation. The first assumption is that the atmosphere is homogeneous and image reflectance is constant, or in part of it since the algorithm allows selection of specific regions for correction. The second assumption is the occurrence of near-zero reflectance values in at least one of the sensor bands, allowing estimation of atmospheric reflectance [22].

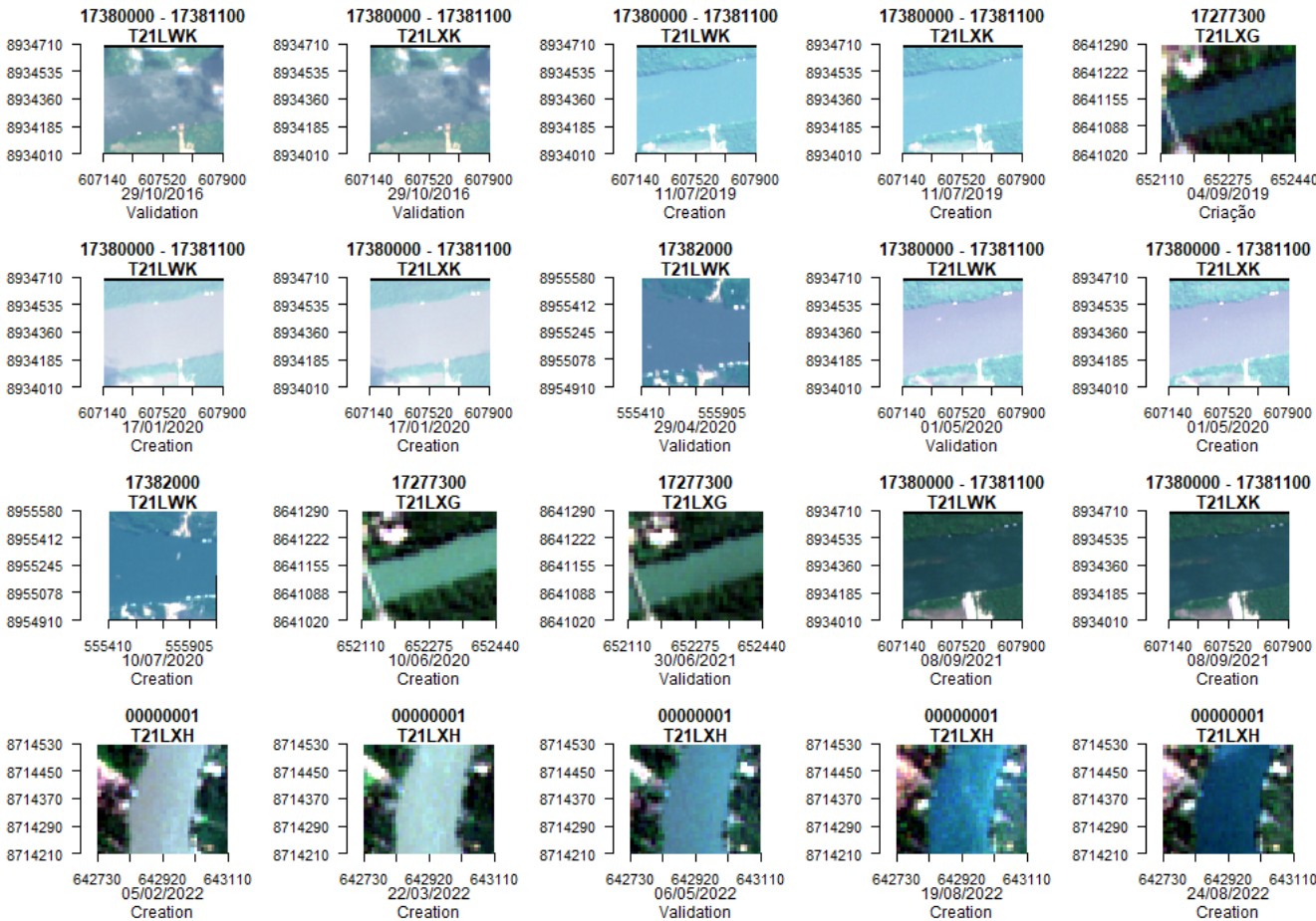

**Figure 3.** Images from Sentinel-2A and B satellites at different sections along the Teles Pires River, Mato Grosso state, Brazil.

Based on these assumptions, top-of-atmosphere reflectance ($\rho_T$) is corrected for gas–water interface [37]. After this, dark-object reflectance ($\rho_{\_dark}$) is defined by using the regression intercept containing the lowest reflectance values in each band. Aerosol thickness is calculated by comparing the $\rho_{\_dark}$ values found in the previous step with aerosol models created using a Look Up Table (LUT). This table is generated using the 6SV model [38] and simulates spectral curves that vary with aerosol optical thickness ($\tau_a$) at 550 nm. For each aerosol model contained in the LUT, the band that provides the smallest $\tau_a$ is chosen. A combination of a band and model with the smallest $\tau_a$ is finally used for calibration, and the parameters required are imported from the LUT.

Another relevant factor in the study of aquatic environments using remote sensing is direct sunlight reflection to the sensor's field of view, known as sun glint. This phenomenon produces anomalies in reflectance capture from water quality parameters by orbital sensors [23]. The ACOLITE processor incorporates a method where a GRS algorithm is used to correct sun glint contamination in SWIR band sensors [27]. The algorithm estimates the bidirectional reflectance distribution factor (BRDF) for air–water interface at the SWIR band and propagates it to visible and near-infrared (NIR) bands [23]. The images selected during verification of intersection with field data were re-downloaded using the Sentinel hub library at processing level 1C. Changes in image acquisition sources occur because the ACOLITE requires auxiliary files for calibration. After atmospheric calibration, images were imported into GEE for data processing.

*2.5. Data Processing*

After importing images to Google Earth Engine (GEE), pixel values were extracted by a semi-automatic process. For all stations, we manually drew a perpendicular line between river banks. From this point on, the entire process was automated. First, a 30-m buffer zone was established on the drawn line, and then a cut was made. To extract only water-related pixels, we applied the modified normalized difference water index (MNDWI), a radiometric index, which replaces the NIR band with the middle-infrared (MIR) band in the Normalized Difference Water Index (NDWI). This alteration provides less interference from anthropic objects and, due to greater energy absorption of MIR by aquatic environments, increases efficiency in delineating water bodies [39]. Like the NDWI, the index varies from $-1$ to 1, with 0 being the threshold between aquatic and non-aquatic environments and positive values corresponding to water. In this way, a vector mask based only on pixels greater than zero was created. A new buffer was established to minimize margin effects, reducing the area by 10 m. For pixel value extraction, a $10 \times 10$ m sample mesh was created, considering spatial resolution of images and avoiding unnecessary interpolations. Since EWI sampling represents average suspended sediment concentration (SSC), pixels contained within the mask were averaged for statistical analysis. Below, Figure 4 describes the entire process, while Figure 5 shows the study design.

All statistics were performed using R 4.0.2 open-source statistical software. First, data normality was verified by the Shapiro–Wilk test at 5% significance ($p = 0.05$). A Spearman correlation matrix was used to assess which variables had the highest correlation with SSC data, and two regression models were later adjusted. The HydroGOF package was used to implement statistical parameters that measure model efficiency [40]. Estimated data were compared with observed data using as parameters mean absolute error (*MAE*; Equation (1)), root mean squared error (*RMSE*; Equation (2)), bias (*BIAS*; Equation (3)), Willmott concordance index (*d*; Equation (4)), the Nash–Sutcliffe efficiency index (*NSE*; Equation (5)), and the mean relative error (*MRE*; Equation (6)).

$$MAE = \frac{1}{N} \sum_{i=1}^{N} |O_i - P_i| \tag{1}$$

$$RMSE = \lceil \frac{1}{N} \sum_{i=1}^{N} (O_i - P_i)^2 \rceil^{0.5} \tag{2}$$

$$BIAS = \frac{1}{N} \sum_{i=1}^{N} (O_i - P_i) \tag{3}$$

$$d = 1 - \lceil \frac{\sum_{i=1}^{N} (P_i - O_i)^2}{\sum_{i=1}^{N} (|P_i - O| + |O_i - O|)^2} \rceil \tag{4}$$

$$NSE = 1 - \frac{\sum_{i=1}^{N} (O_i - P_i)^2}{\sum_{i=1}^{N} (O_i - O)^2} \tag{5}$$

$$MRE = \frac{1}{N} \left( \sum_{i=1}^{N} \frac{(P_i - O_i)}{O_i} \times 100 \right) \tag{6}$$

where $P_i$ is the estimated suspended sediment concentration (SSC) estimated (milligrams (mg)/liter (L)), $O_i$ is the observed SSC (mg/L), O is the average of the observed SSC values (mg/L), and N is the number of values in the sample. MAE and RMSE were used to evaluate variation of errors in SSC estimates for spectral bands B3 and B4. Both indices showed perfect model fit when their values were equal to zero [41]. Bias was used to verify whether the model underestimates or overestimates. Positive values indicate underestimation, while negative values mean overestimation. The Willmott concordance index (d) was applied to evaluate model prediction performance with values ranging from 0 to 1. Here,

a value of d = 1 indicates perfect agreement. Model performance was established using the Nash–Sutcliffe efficiency (NSE) index, where NSE = 1 means a perfect fit of data by the model, NSE > 0.75 suggests an adequate model, NSE between 0.36 and 0.75 indicates a satisfactory model, and NSES < 0.36 indicates an unsatisfactory model [42]. The mean relative error compares the observed values with the estimated ones and expresses the differences in percentages. This parameter is useful when dealing with values with high or low scales.

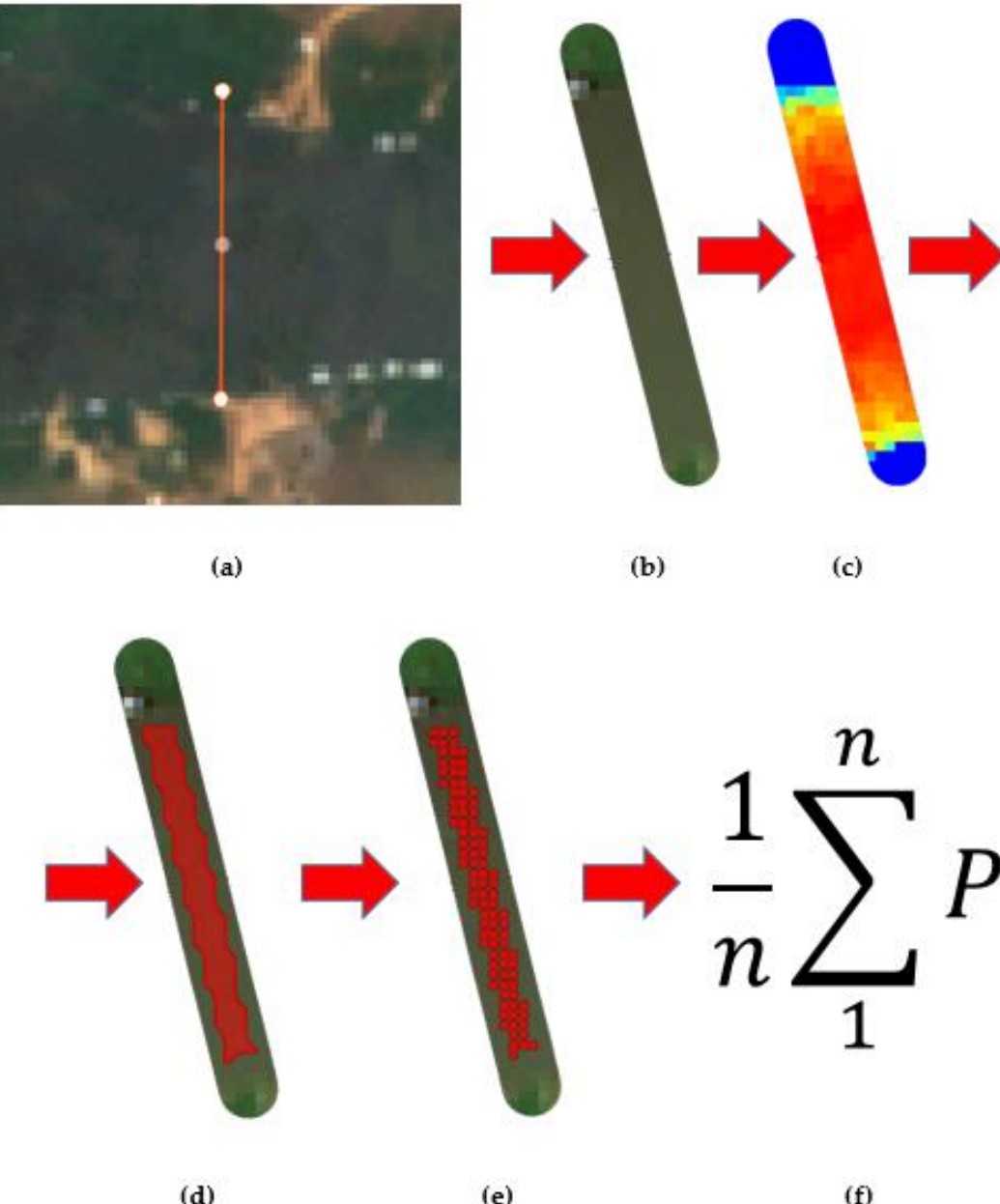

**Figure 4.** Data processing with (**a**) vector drawn manually from one bank to the other, (**b**) 30-m (m) buffer application from the line and cut, (**c**) MNDWI index application to extract only the aquatic environment, (**d**) 10-m buffer application, (**e**) Sample grid of 10 × 10 m, and (**f**) average of values captured by the sampling grid.

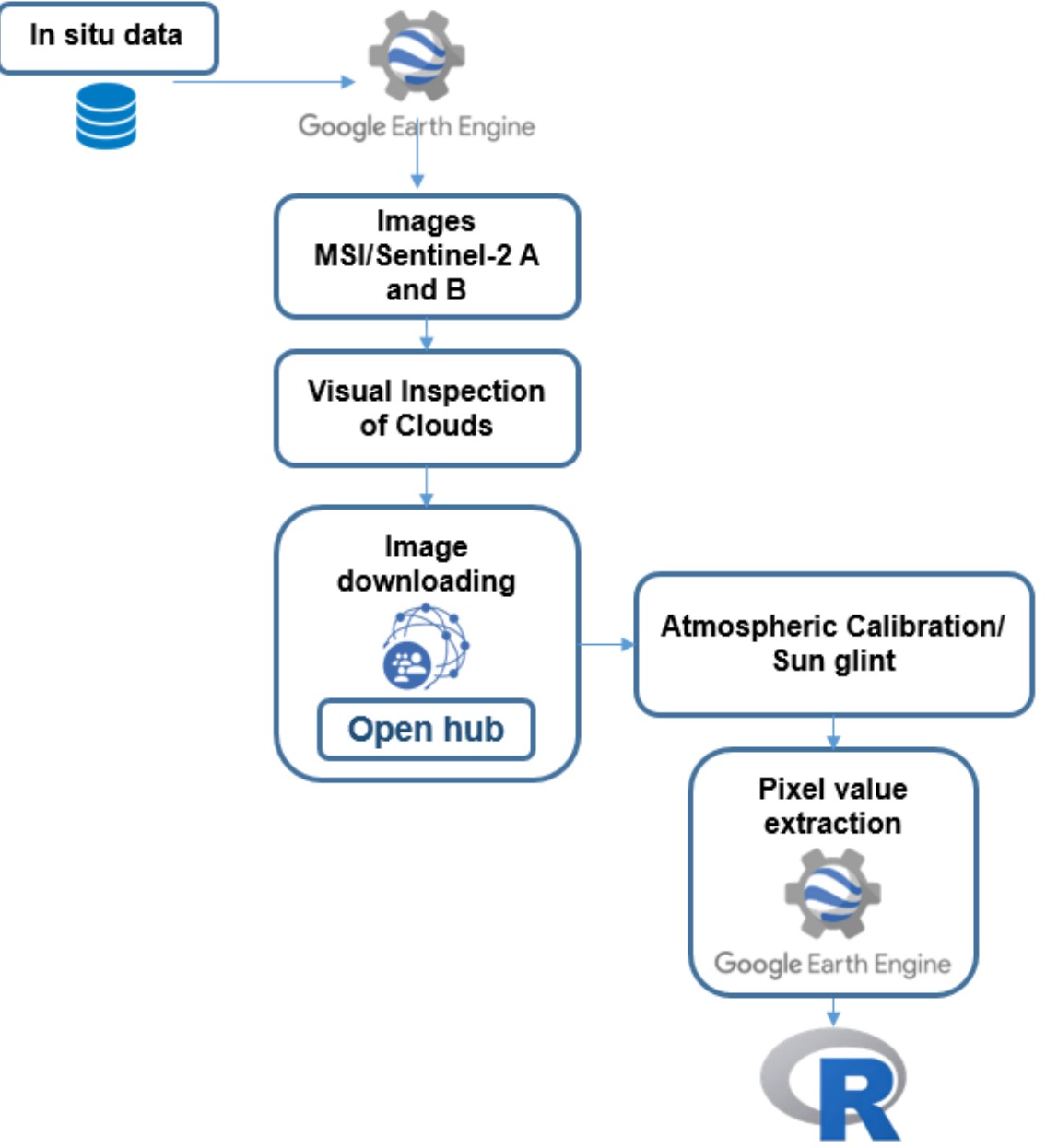

**Figure 5.** Flowchart showing the methods used, from SSC data, satellite image selection, atmospheric calibration, and data processing, using a semi-automatic method.

## 3. Results

Table 2 shows the results from all processing and indicates which data (spectral bands and radiometric indices) were used to create and validate suspended sediment concentration (SSC) estimation models. The values of spectral bands B3 and B4 and the Normalized Difference Vegetation Index (NDVI) showed normality with $p = 0.105$, $p = 0.283$, and $p = 0.059$, whereas the in-situ SSC, B8, and Normalized Difference Water Index (NDWI) showed no normality with $p = 0.002$, $p = 0.001$, and $p = 0.024$ (Table 3). Of the analyzed variables, band B4 had the highest correlation coefficient with Rho = 0.726, and B3 had Rho = 0.677 (Table 3). Therefore, there is a strong correlation between the red and green spectrums and SSC. SCC was significantly different for B3, B4, and B8 spectral bands but not for NDWI and NDVI (Table 3). Two regression models were fit to the B3 and B4 bands, which were linear (Figure 6) and exponential (Figure 7).

**Table 2.** Data from Agência Nacional de Águas' (ANA's) hydro-sedimentological stations specified as codes, sediment collection dates, and images with their function of use in the models, reflectance values from different spectral bands, radiometric normalized difference indexes of the Sentinel satellite imagery for water (NDWI) and vegetation (NDVI), and suspended sediment concentrations (SSC).

| Code | Date | Function | Spectral Band B3 | Spectral Band B4 | Spectral Band B8 | NDWI | NDVI | SSC |
|---|---|---|---|---|---|---|---|---|
| 17381100 | 29 October 2016 | Validation | 0.053753 | 0.038952 | 0.034259 | 0.250606 | −0.08914 | 17.55 |
| 17381100 | 29 October 2016 | Validation | 0.053747 | 0.038953 | 0.034259 | 0.250555 | −0.08915 | 17.55 |
| 17380000 | 11 July 2019 | Creation | 0.033997 | 0.018504 | 0.015644 | 0.37387 | −0.08958 | 5.10 |
| 17380000 | 11 July 2019 | Creation | 0.033996 | 0.018503 | 0.015605 | 0.374827 | −0.09063 | 5.10 |
| 17277300 | 4 September 2019 | Creation | 0.067924 | 0.053814 | 0.052215 | 0.130831 | −0.0152 | 8.91 |
| 17381100 | 17 January 2020 | Creation | 0.04712 | 0.04482 | 0.067682 | −0.17882 | 0.2029 | 9.82 |
| 17381100 | 17 January 2020 | Creation | 0.047095 | 0.044796 | 0.06767 | −0.17899 | 0.20307 | 9.82 |
| 17382000 | 29 April 2020 | Validation | 0.03022 | 0.02803 | 0.013714 | 0.382123 | −0.34932 | 9.39 |
| 17381100 | 1 May 2020 | Creation | 0.029089 | 0.025905 | 0.014683 | 0.327065 | −0.27623 | 6.74 |
| 17381100 | 1 May 2020 | Validation | 0.029093 | 0.02591 | 0,.014682 | 0.327161 | −0.27635 | 6.74 |
| 17277300 | 10 June 2020 | Creation | 0.051608 | 0.039467 | 0.017445 | 0.495145 | −0.38752 | 14.42 |
| 17382000 | 10 July 2020 | Creation | 0.033754 | 0.018002 | 0.017074 | 0.329243 | −0.02757 | 5.05 |
| 17277300 | 30 June 2021 | Validation | 0.050261 | 0.038579 | 0.02081 | 0.414598 | −0.29942 | 5.00 |
| 17381100 | 8 September 2021 | Creation | 0.027345 | 0.010911 | 0.021023 | 0.133084 | 0.329158 | 6.28 |
| 17381100 | 8 September 2021 | Creation | 0.027333 | 0.010893 | 0.021005 | 0.133293 | 0.329555 | 6.28 |
| 00000001 | 5 February 2022 | Creation | 0.055942 | 0.060056 | 0.029518 | 0.309608 | −0.34133 | 17.42 |
| 00000001 | 22 March 2022 | Creation | 0.05877 | 0.063464 | 0.046997 | 0.11156 | −0.14925 | 24.37 |
| 00000001 | 6 May 2022 | Validation | 0.048097 | 0.03775 | 0.023068 | 0.352535 | −0.24231 | 11.63 |
| 00000001 | 19 August 2022 | Creation | 0.037834 | 0.023956 | 0.041534 | −0.04737 | 0.269369 | 6.53 |
| 00000001 | 24 August 2022 | Creation | 0.036101 | 0.016822 | 0.013989 | 0.443149 | −0.09479 | 5.58 |

**Table 3.** Tests for normality, correlation, and significant differences [a] of suspended sediment concentrations (SSC) for the spectral bands and radiometric indices selected.

| Element | Normality (Shapiro–Wilk) | Correlation (*Rho*) | Significance (*p*-Value) |
|---|---|---|---|
| Suspended sediment concentration | 0.002 ** | - | - |
| Spectral Band | | | |
| B3 | 0.105 * | 0.677 | 0.001 ** |
| B4 | 0.283 ** | 0.760 | 0.001 *** |
| B8 | 0.001 ** | 0.359 | 0.017 ** |
| Radiometric Index | | | |
| Normalized Difference Water Index (NDWI) | 0.024 * | −0.097 | 0.683 |
| Normalized Difference Vegetation Index (NDVI) | 0.059 * | −0.276 | 0.239 |

[a] Differences significant at a confidence level of 0.1 *, 0.05 **, and 0.01 ***, respectively.

In both models, band B4 had the best coefficients of determination ($R^2$), which were 0.5849 for the linear and 0.7883 for the exponential; therefore, the exponential model provides the best fit for band B4. In band B3, the same behavior occurred, with the linear fit having an $R^2$ of 0.4972 and the exponential an $R^2$ of 0.7003. Table 4 shows the evaluation of the linear and exponential models for bands B3 and B4. For band B4, the exponential model showed the lowest mean absolute error (MAE) and root mean square (RMSE) indices. For band B3, the MAE index indicated smaller errors for the exponential model, while the RMSE index indicated the linear model as more accurate. The BIAS showed that only the linear model with B3 underestimated the suspended sediment concentration (SSC). The Willmott concordance index (d) found the highest concordance for the estimator for the B4 band, with the exponential model presenting a value closer to 1. The same combination also presented the best result for the Nash–Sutcliffe efficiency index (NSE) (0.75), which indicated that the model was satisfactory. For the two bands studied, the exponential model generated the lowest mean relative errors; in particular, band B4 showed the best results

for this parameter in both models. Figure 8 expresses the comparison between observed and estimated data for the validation set.

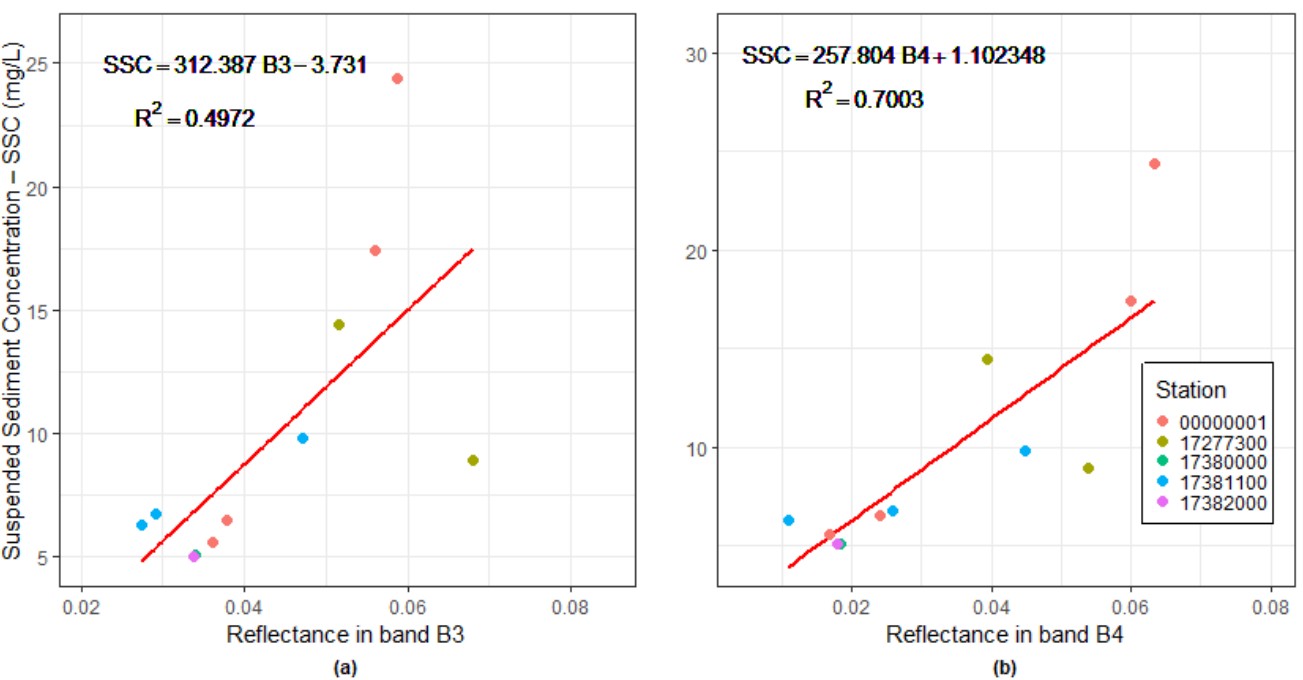

**Figure 6.** Linear models: with (**a**) linear fit between B3 band (green) and suspended sediment concentration (SSC) and (**b**) linear fit between B4 band (red) and SSC.

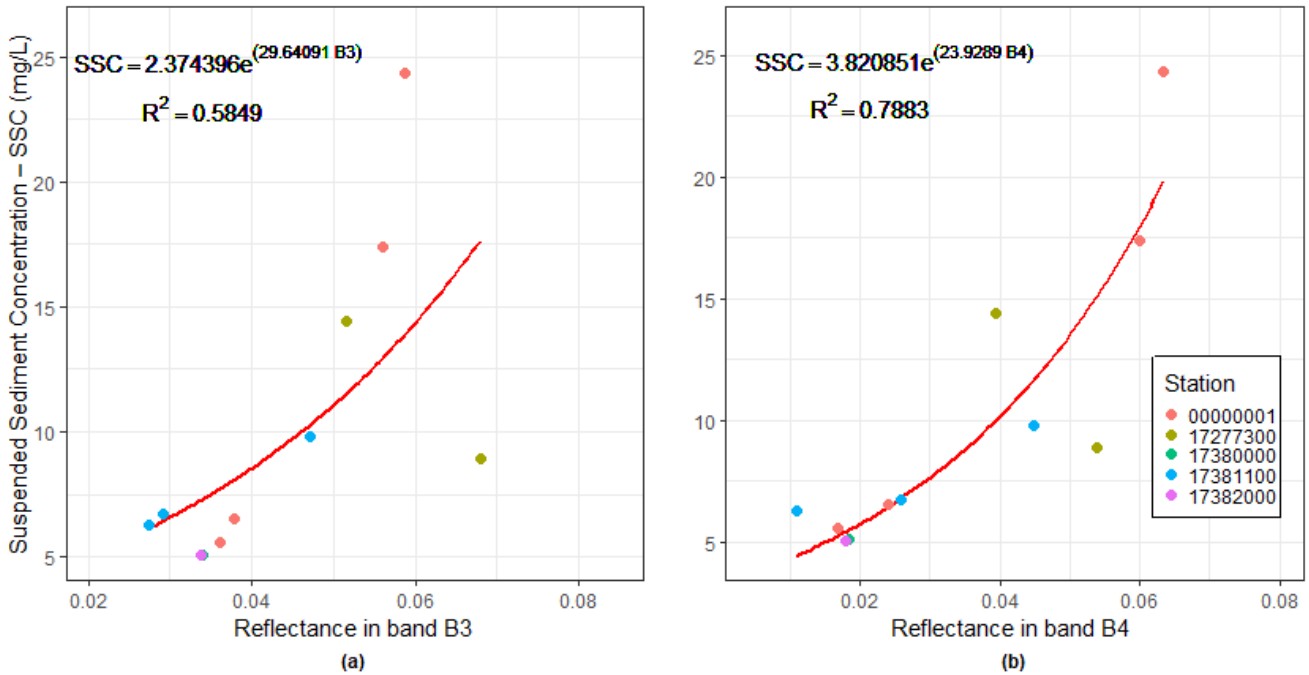

**Figure 7.** Linear and exponential models for fit between (**a**) band B3 (green) and suspended sediment concentration (SSC) and between (**b**) band B4 (red) and SSC.

**Table 4.** Statistical evaluation of the models adjusted for spectral bands and suspended sediment concentrations (SSC).

| | Spectral Band B3 | | Spectral Band B4 | |
|---|---|---|---|---|
| **Statistical Parameter** | **Linear Model** | **Exponential Model** | **Linear Model** | **Exponential Model** |
| Mean absolute error (MAE) (mg/L) | 2.825193 | 2.7046560 | 2.251899 | 1.8859822 |
| Root mean squared error (RMSE) (mg/L) | 3.881253 | 4.1712430 | 2.996691 | 2.7348836 |
| BIAS | $-3.55 \times 10^{-15}$ | 0.5354025 | $-8.2462 \times 10^{-16}$ | $-0.3289959$ |
| Willmott's concordance (d) index | 0.8011445 | 0.7504934 | 0.9041224 | 0.9139091 |
| Nash–Sutcliffe efficiency (NSE) index | 0.497190 | 0.4192480 | 0.7002607 | 0.7503466 |
| Mean relative error (%) | 30.48 | 26.15 | 23.54 | 18.15% |

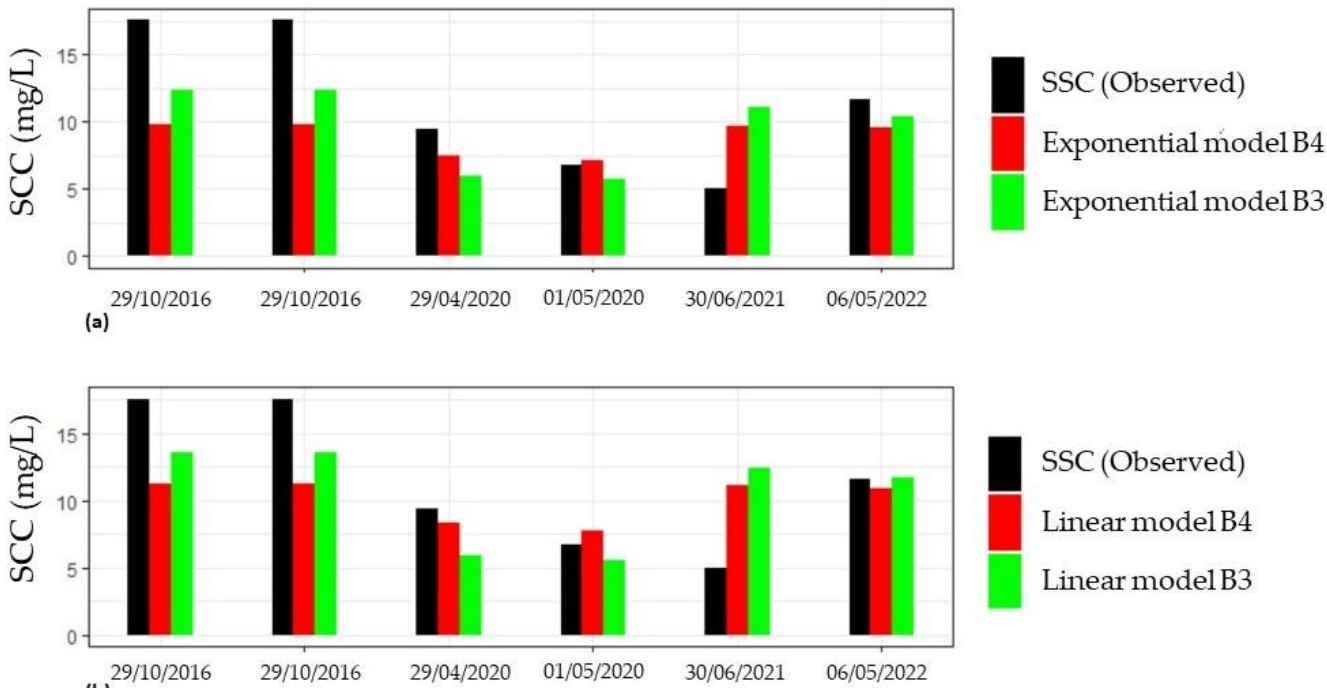

**Figure 8.** Comparison between exponential and linear models: (**a**) Exponential models B3 and B4 (**b**) Linear models B3 and B4.

## 4. Discussion

### 4.1. Comparisons to Previous Research

Data on in situ suspended sediment concentration (SSC), Normalized Difference Water Index (NDWI), and spectral band B8 were not significant for a normal distribution of the data. This may be due to the limitation of observations to the intersection of in situ/satellite collections. Both spectral bands B3 and B4 had significantly normal data distributions. The latter spectral band B4 was the best estimator for both the linear and exponential models. This result is in line with that of Marinho et al. [19], who found strong relationships between SSC and spectral band B4 (red) of the Multi-Spectral Instrument (MSI) sensor used for remote sensing of the Negro River in the northern Amazon forest in Brazil. Moreover, Santos et al. [20] reached the same conclusion for the Purus River in Brazil's eastern Amazon forest using the MODIS sensor. This was also confirmed by Lobo et al. [15] for the Tapajós River in the eastern part of the Brazilian Amazon using the Landsat-MSS/TM/OLI sensors.

These studies corroborated our results, where the spectral band B4 (red) explained suspended sediments better, even at low concentrations. However, Jensen [34] makes

a point that more collecting data on chlorophyll content may be needed to improve the accuracy of the models that were used in all of these studies. This author explained that such behavior is due to a link between reflectance shift from red and infrared to green with chlorophyll content in the aquatic environment at low sediment concentrations. Thus, the hypothesis that the Teles Pires River has low chlorophyll contents would explain why band B4 (red) was more efficient in estimating SSC, even at low suspended sediment concentrations. However, to validate this claim, chlorophyll concentrations along the Teles Pires River should be measured in the future.

Radiometric indexes had no correlation with SSC data, especially the Normalized Difference Water Index (NDWI), contradicting the findings of Simões et al. [17]. This difference may have been due to the use of images from days after and before the days of collection in the field. Another hypothesis was launched by McFeeters [36], who had already warned that the broadband aspect of NDWI would increase efficiency in estimating general turbidity. However, such efficiency would be lost when the index is related to isolated variables, such as chlorophyll and suspended sediments. Krug and Noernberg [43] applied NDWI to determine bathymetry and detected that the index also suffers interference from depth variations. Since Simões et al. [17] used the index in two fixed sections, depth was constant, and thus NDWI values were mostly affected by changes in suspended material, which could explain the divergence between the results. The weak correlation of the Normalized Difference Vegetation Index (NDVI) with suspended sediment concentration (SSC) can be explained by very low reflectance values in spectral bands B4 and B8. The genesis of the index indicates that satisfactory results are linked to significant differences between the reflectance values of study objects [35]. SSC in the Teles Pires River provides small reflectance in the red region. Such spectral behavior, combined with high infrared absorption by water, makes the index ineffective in detecting sediments.

The mathematical model that presented the best fit and accuracy was the exponential model, which presented the smallest errors. The combination of the exponential model with the reflectance of the B4 band showed the best results in all the statistical indices that we evaluated. Some stretches studied are at overlapping points of satellite images, which provided two images for the same day, reinforcing the model. The average SSC for the Teles Pires River is approximately 11 mg/L, and the SSC values used to create the model ranged from 5.05 to 24.37 mg/L, demonstrating the feasibility of the model for this water body. When evaluating the application of the model to the validation dataset, the model was able to detect the variation in sediment concentration along the section. Figure 9 demonstrates the potential of this methodology/model in the spatialization of information, as it converts specific SSC information into regional SSC information.

The exponential model had the best fit and accuracy, showing the lowest errors regarding the observed data. Some sections are at overlapping points of satellite images, which provided two images for the same day, reinforcing the test of the model. The average SSC for the Teles Pires River was about 11 milligrams (mg)/liter (L), with values used for model creation varying from 5.05 to 24.37 mg/L. Therefore, the model is feasible for sampling in this water body. No measurements could be evaluated for the last quarter of 2019 or 2020. This situation is related to the rainy season, during which clouds are common in satellite images for the study region. This difficulty can be minimized by using other orbital sensors, increasing the likelihood of capturing images without clouds, as was done in other studies [15,18]. Another solution was proposed by researchers who used unmanned aircraft with onboard spectral sensors during field collections [44,45]. When using the proper method, this approach would reduce impacts from the atmosphere and give some freedom for planning field sampling.

The Google Earth Engine (GEE) tool proved to be flexible in processing images. By applying this tool, several process settings are allowed without requiring an excessive amount of labor. With a few lines of code, we could quickly and efficiently verify the adequacy, occurrence, or lack of occurrence of any image without clouds or noise for the

53 dates of our field sampling. The processing power and availability of remote sensing data make GEE a robust tool for studying water quality parameters.

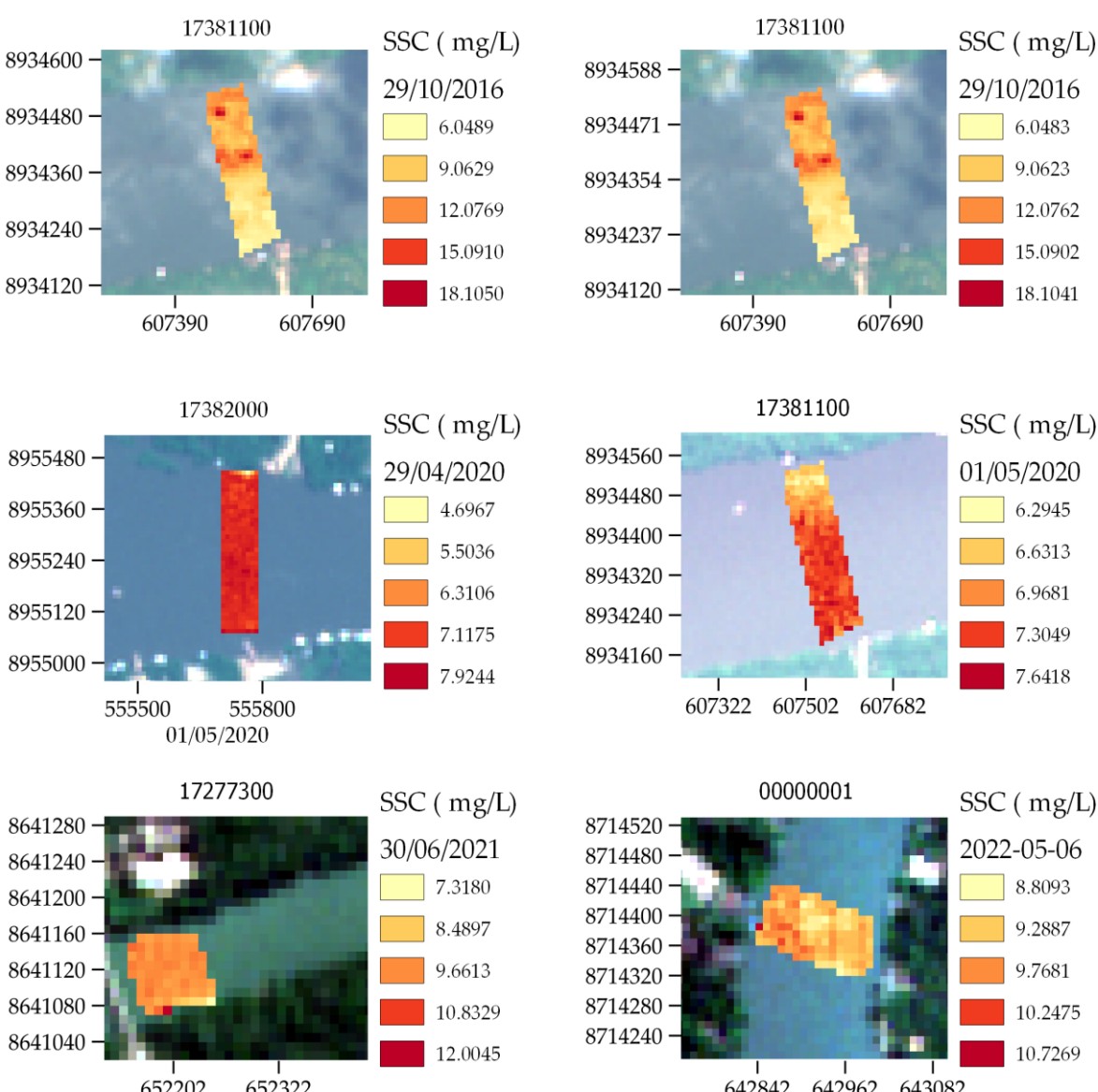

**Figure 9.** Application of the B4 exponential model on the set of images for validation.

### 4.2. Sustainable Agricultural Development Implications and Public Policy Recommendations

In addition to impacting the useful life of the reservoirs, estimating and/or monitoring suspended sediment concentration (SSC) sheds light on the erosive processes in hydrographic basins. Erosive processes can generate several consequences and vary according to the characteristics of the basin, such as vegetative cover, topography, precipitation regime, and soil properties [46]. In particular, water erosion leads to soil degradation by reducing nutrients and organic matter and favors the transport of fertilizers and pesticides, directly impacting economic activities in these basins [47]. The intensification of erosion can cause damage to the environment and agricultural production affecting food security [48]. Erosion is also intensified in hydrographic basins with a predominance of cultivated or deforested areas [49]. Thus, obtaining SSC data and, consequently, erosion data are essential for the proper management of soil and water in such watersheds.

Recent measurements of suspended sediment concentration (SSC) in the Teles Pires river basin show that SSC is higher in those parts of the basin that have more agricultural

land and urban areas [50]. Therefore, it is important to develop public policies that can engage agricultural producers and urban municipalities in the proposition, evaluation, and codification of such policies to encourage the reduction of SSC in the Teles Pires River and other rivers in Brazil. Despite the importance of monitoring solids in water resources, public policies focused on SSC have been limited to monitoring the areas drained from the dams of Hydroelectric Projects. This policy was established by ANA/ANEEL resolution n° 03/2010. However, there is no specific limit to the concentration of solids in watercourses, possibly due to the enormous cost of the necessary measures, a more detailed understanding of erosion processes, and due to poor soil management and use.

Our research raises the possibility of establishing models for estimating the concentration of suspended sediments (SSC) with greater spatiality and temporality. This estimation is possible throughout the water body studied and even over time. Therefore, it is possible to determine the causes and effects of erosion along water resources. With more in-depth studies in the future, different drivers of erosion can be more clearly connected to how they contribute to soil loss and sediment suspension in water bodies. For example, previous research verified how different tributaries contribute to erosion in urban reservoirs located in Campo Grande, midwest Brazil [51]. Understanding where to limit erosion can not only benefit urban areas but also reduce environmental impacts in agricultural areas in Brazil and beyond.

## 5. Conclusions

Remote sensing is an efficient tool to estimate suspended sediment concentration (SSC) since, even with low sediment concentrations from samples taken from the Teles Pires River, the method can detect SSC variations in the sections studied. The isolated bands provided stronger correlations when compared with radiometric indices, and the B4 band (red) was the best SSC estimator. When the exponential model was used, this band obtained the best results with a coefficient of determination equal to $R^2 = 0.7883$ and the Nash-Sutcliffe index with ENS = 0.7503, indicating the feasibility of the model for the Teles Pires River. Google Earth Engine (GEE) has clearly made the exploratory analysis more agile and efficient for image acquisition and method application. For future work, the use of sensor harmonization products such as The Harmonized Landsat Sentinel-2 (HLS) may increase the probability of occurrence of orbital images on the same day of in situ collections, in addition to minimizing efforts in pre-processing. Another approach that could involve this type of study would be the execution of punctual samples of SSC, allowing for the segmentation of variation of the SSC along the section. Moreover, the application of sediment source fingerprint (SSF), associated with land use analysis, can also be used to develop public policies to encourage more sustainable agricultural development.

**Author Contributions:** Data collection, writing, methodology, formal analysis, figures and tables—R.S.D.P. and F.T.d.A.; review, editing, supervision—F.T.d.A.; review, editing, supervision, and financial support—A.P.d.S., A.K.H., and D.C.d.A.; data collection—J.W.d.S.A. and C.C.M. All authors have read and agreed to the published version of the manuscript.

**Funding:** This study was financed by the Coordenação de Aperfeiçoamento de Pessoal de Nível Superior—Brasil (CAPES) and the Agência Nacional de Águas e Saneamento Básico (ANA), Finance Code—001 and Process 88887.144957/2017-00. The authors also thank the Coordination for the Improvement of Higher Education Personnel—Brazil (CAPES) for supporting the graduate scholarship and the National Council of Scientific and Technological Development (CNPq) for supporting scientific initiation scholarships, as well as the AgriScience project.

**Institutional Review Board Statement:** Not applicable.

**Informed Consent Statement:** Not applicable.

**Data Availability Statement:** Study data can be obtained by request to the corresponding author or the first author via e-mail. It is not available on the website as the research project is still under development.

**Acknowledgments:** The authors thank the Coordenação de Aperfeiçoamento de Pessoal de Nível Superior—Brasil (CAPES) and the Agência Nacional de Águas e Saneamento Básico (ANA), Finance Code—001 and Process 88887.144957/2017-00. The authors also thank the Coordination for the Improvement of Higher Education Personnel—Brazil (CAPES) for supporting the graduate scholarship and the National Council of Scientific and Technological Development (CNPq) for the support with scientific initiation scholarships, as well as the AgriScience Project. The authors also thank all the students and professors of the Tecnologia em Recursos Hídricos no Centro-Oeste research group (dgp.cnpq.br/dgp/espelhogrupo/2399343537529589, accessed 29 January 2023).

**Conflicts of Interest:** The authors declare no conflict of interest. Supporting entities had no role in the design of the study; in the collection, analyses, or interpretation of data; in the writing of the manuscript, or in the decision to publish the results.

## Appendix A

**Table A1.** Data used in the study, such as the code of the field measurement station (Code) and its description (Name), the concomitant date of the field collection and the satellite image (Date), and the concentration of suspended sediments (SSC).

| Code | Name | Date | SSC |
|---|---|---|---|
| 17307000 | HEP Colíder Pesqueiro do Gil | 16 April 2019 | 15.40 |
| 17307000 | HEP Colíder Pesqueiro do Gil | 17 July 2019 | 2.40 |
| 17307000 | HEP Colíder Pesqueiro do Gil | 22 June 2021 | 3.30 |
| 17307000 | HEP Colíder Pesqueiro do Gil | 24 September 2021 | 2.00 |
| 17390100 | HEP São Manuel downstream 1 | 5 February 2016 | 24.24 |
| 17390100 | HEP São Manuel downstream 1 | 25 June 2016 | 12.61 |
| 17390100 | HEP São Manuel downstream 1 | 2 September 2016 | 10.78 |
| 17390100 | HEP São Manuel downstream 1 | 15 December 2016 | 25.64 |
| 17390100 | HEP São Manuel downstream 1 | 15 February 2017 | 25.43 |
| 17390100 | HEP São Manuel downstream 1 | 22 May 2017 | 10.96 |
| 17390100 | HEP São Manuel downstream 1 | 7 August 2017 | 8.95 |
| 17390100 | HEP São Manuel downstream 1 | 6 November 2017 | 9.95 |
| 17390100 | HEP São Manuel downstream 1 | 31 January 2018 | 14.11 |
| 17390100 | HEP São Manuel downstream 1 | 28 June 2018 | 12.55 |
| 17390100 | HEP São Manuel downstream 1 | 3 September 2018 | 9.41 |
| 17390100 | HEP São Manuel downstream 1 | 5 November 2018 | 10.35 |
| 17390100 | HEP São Manuel downstream 1 | 20 February 2019 | 26.98 |
| 17390100 | HEP São Manuel downstream 1 | 23 May 2019 | 11.43 |
| 17390100 | HEP São Manuel downstream 1 | 21 August 2019 | 9.18 |
| 17390100 | HEP São Manuel downstream 1 | 20 November 2019 | 14.56 |
| 17390100 | HEP São Manuel downstream 1 | 4 February 2020 | 13.78 |
| 17390100 | HEP São Manuel downstream 1 | 8 October 2020 | 11.21 |
| 17390100 | HEP São Manuel downstream 1 | 26 November 2020 | 11.40 |
| 17390100 | HEP São Manuel downstream 1 | 9 March 2021 | 10.11 |
| 17390100 | HEP São Manuel downstream 1 | 23 May 2021 | 8.75 |
| 17390100 | HEP São Manuel downstream 1 | 2 July 2021 | 9.25 |
| 17390100 | HEP São Manuel downstream 1 | 2 December 2021 | 9.51 |
| 17390100 | HEP São Manuel downstream 1 | 10 December 2021 | 10.60 |

**Table A1.** *Cont.*

| Code | Name | Date | SSC |
|---|---|---|---|
| 17277300 | HEP Sinop upstream 1 | 5 March 2019 | 36.10 |
| 17277300 | HEP Sinop upstream 1 | 13 June 2019 | 10.66 |
| 17277300 | HEP Sinop upstream 1 | 4 September 2019 | 8.91 |
| 17277300 | HEP Sinop upstream 1 | 11 December 2019 | 42.17 |
| 17277300 | HEP Sinop upstream 1 | 18 March 2020 | 34.59 |
| 17277300 | HEP Sinop upstream 1 | 10 June 2020 | 14.42 |
| 17277300 | HEP Sinop upstream 1 | 9 September 2020 | 6.77 |
| 17277300 | HEP Sinop upstream 1 | 15 October 2021 | 2.00 |
| 17277300 | HEP Sinop upstream 1 | 30 June 2021 | 5.00 |
| 17277300 | HEP Sinop upstream 1 | 4 December 2021 | 12.00 |
| 17277300 | HEP Sinop upstream 1 | 2 April 2021 | 25.56 |
| 17277300 | HEP Sinop upstream 1 | 12 December 2020 | 17.36 |
| 17381100 | HEP Teles Pires upstream 2 | 3 February 2016 | 29.22 |
| 17381100 | HEP Teles Pires upstream 2 | 10 May 2016 | 16.97 |
| 17381100 | HEP Teles Pires upstream 2 | 19 July 2016 | 13.31 |
| 17381100 | HEP Teles Pires upstream 2 | 29 October 2016 | 17.55 |
| 17381100 | HEP Teles Pires upstream 2 | 25 January 2017 | 19.93 |
| 17381100 | HEP Teles Pires upstream 2 | 17 April 2017 | 13.77 |
| 17381100 | HEP Teles Pires upstream 2 | 27 July 2017 | 12.30 |
| 17381100 | HEP Teles Pires upstream 2 | 30 October 2017 | 12.01 |
| 17381100 | HEP Teles Pires upstream 2 | 24 January 2018 | 17.81 |
| 17381100 | HEP Teles Pires upstream 2 | 15 April 2018 | 12.15 |
| 17381100 | HEP Teles Pires upstream 2 | 5 July 2018 | 9.60 |
| 17381100 | HEP Teles Pires upstream 2 | 31 October 2018 | 14.31 |
| 17381100 | HEP Teles Pires upstream 2 | 18 January 2019 | 9.43 |
| 17381100 | HEP Teles Pires upstream 2 | 24 April 2019 | 4.91 |
| 17381100 | HEP Teles Pires upstream 2 | 20 July 2019 | 6.29 |
| 17381100 | HEP Teles Pires upstream 2 | 31 October 2019 | 6.52 |
| 17381100 | HEP Teles Pires upstream 2 | 17 January 2020 | 9.82 |
| 17381100 | HEP Teles Pires upstream 2 | 1 May 2020 | 6.74 |
| 17381100 | HEP Teles Pires upstream 2 | 11 July 2020 | 56.40 |
| 17381100 | HEP Teles Pires upstream 2 | 16 September 2020 | 5.64 |
| 17381100 | HEP Teles Pires upstream 2 | 30 January 2021 | 12.66 |
| 17381100 | HEP Teles Pires upstream 2 | 13 May 2021 | 9.64 |
| 17381100 | HEP Teles Pires upstream 2 | 8 September 2021 | 6.28 |
| 17381100 | HEP Teles Pires upstream 2 | 24 November 2021 | 10.96 |
| 17381100 | HEP Teles Pires upstream 2 | 22 October 2020 | 4.28 |
| 17382000 | HEP Teles Pires upstream 1 | 3 February 2016 | 25.13 |
| 17382000 | HEP Teles Pires upstream 1 | 13 May 2016 | 17.26 |
| 17382000 | HEP Teles Pires upstream 1 | 15 July 2016 | 12.79 |

**Table A1.** *Cont.*

| Code | Name | Date | SSC |
|---|---|---|---|
| 17382000 | HEP Teles Pires upstream 1 | 21 October 2016 | 14.04 |
| 17382000 | HEP Teles Pires upstream 1 | 28 January 2017 | 18.49 |
| 17382000 | HEP Teles Pires upstream 1 | 15 April 2017 | 16.44 |
| 17382000 | HEP Teles Pires upstream 1 | 24 July 2017 | 10.39 |
| 17382000 | HEP Teles Pires upstream 1 | 28 October 2017 | 11.43 |
| 17382000 | HEP Teles Pires upstream 1 | 26 January 2018 | 17.08 |
| 17382000 | HEP Teles Pires upstream 1 | 16 April 2018 | 15.50 |
| 17382000 | HEP Teles Pires upstream 1 | 7 July 2018 | 10.04 |
| 17382000 | HEP Teles Pires upstream 1 | 2 November 2018 | 18.02 |
| 17382000 | HEP Teles Pires upstream 1 | 17 January 2019 | 11.43 |
| 17382000 | HEP Teles Pires upstream 1 | 23 April 2019 | 7.18 |
| 17382000 | HEP Teles Pires upstream 1 | 23 July 2019 | 5.97 |
| 17382000 | HEP Teles Pires upstream 1 | 1 November 2019 | 3.75 |
| 17382000 | HEP Teles Pires upstream 1 | 21 January 2020 | 5.87 |
| 17382000 | HEP Teles Pires upstream 1 | 29 April 2020 | 9.39 |
| 17382000 | HEP Teles Pires upstream 1 | 10 July 2020 | 5.05 |
| 17382000 | HEP Teles Pires upstream 1 | 17 September 2020 | 3.12 |
| 17382000 | HEP Teles Pires upstream 1 | 22 October 2020 | 3.48 |
| 17382000 | HEP Teles Pires upstream 1 | 30 January 2021 | 12.92 |
| 17384200 | HEP Teles Pires downstream | 11 February 2016 | 10.35 |
| 17384200 | HEP Teles Pires downstream | 26 June 2016 | 10.85 |
| 17384200 | HEP Teles Pires downstream | 20 July 2016 | 11.62 |
| 17384200 | HEP Teles Pires downstream | 2 November 2016 | 11.42 |
| 17384200 | HEP Teles Pires downstream | 27 January 2017 | 13.50 |
| 17384200 | HEP Teles Pires downstream | 18 April 2017 | 10.41 |
| 17384200 | HEP Teles Pires downstream | 28 July 2017 | 8.59 |
| 17384200 | HEP Teles Pires downstream | 3 November 2017 | 9.26 |
| 17384200 | HEP Teles Pires downstream | 29 January 2018 | 10.02 |
| 17384200 | HEP Teles Pires downstream | 17 April 2018 | 8.93 |
| 17384200 | HEP Teles Pires downstream | 3 July 2018 | 11.30 |
| 17384200 | HEP Teles Pires downstream | 1 November 2018 | 12.66 |
| 17384200 | HEP Teles Pires downstream | 16 January 2019 | 2.63 |
| 17384200 | HEP Teles Pires downstream | 22 April 2019 | 4.17 |
| 17384200 | HEP Teles Pires downstream | 19 July 2019 | 11.88 |
| 17384200 | HEP Teles Pires downstream | 1 November 2019 | 2.64 |
| 17384200 | HEP Teles Pires downstream | 20 January 2020 | 3.27 |
| 17384200 | HEP Teles Pires downstream | 28 April 2020 | 4.19 |
| 17384200 | HEP Teles Pires downstream | 9 July 2020 | 5.88 |
| 17384200 | HEP Teles Pires downstream | 18 September 2020 | 2.91 |

**Table A1.** *Cont.*

| Code | Name | Date | SSC |
|---|---|---|---|
| 17384200 | HEP Teles Pires downstream | 21 October 2020 | 0.45 |
| 17384200 | HEP Teles Pires downstream | 1 February 2021 | 6.20 |
| 17380000 | Downstream the mouth of Peixoto de Azevedo | 24 August 2016 | 5.80 |
| 17380000 | Downstream the mouth of Peixoto de Azevedo | 30 November 2016 | 19.90 |
| 17380000 | Downstream the mouth of Peixoto de Azevedo | 2 May 2017 | 8.00 |
| 17380000 | Downstream the mouth of Peixoto de Azevedo | 24 July 2017 | 3.10 |
| 17380000 | Downstream the mouth of Peixoto de Azevedo | 31 October 2017 | 6.10 |
| 17380000 | Downstream the mouth of Peixoto de Azevedo | 20 July 2018 | 4.60 |
| 17380000 | Downstream the mouth of Peixoto de Azevedo | 25 October 2018 | 9.80 |
| 17380000 | Downstream the mouth of Peixoto de Azevedo | 6 May 2019 | 13.30 |
| 17380000 | Downstream the mouth of Peixoto de Azevedo | 11 July 2019 | 5.10 |
| 17380000 | Downstream the mouth of Peixoto de Azevedo | 30 October 2019 | 13.80 |
| 17380000 | Downstream the mouth of Peixoto de Azevedo | 16 November 2021 | 26.80 |
| 17380000 | Downstream the mouth of Peixoto de Azevedo | 27 April 2022 | 16.30 |
| 17380000 | Downstream the mouth of Peixoto de Azevedo | 26 July 2022 | 8.90 |
| 00000001 | Section Curio | 19 August 2022 | 6.53 |
| 00000001 | Section Curio | 5 February 2022 | 17.42 |
| 00000001 | Section Curio | 24 August 2022 | 10.74 |
| 00000001 | Section Curio | 22 March 2022 | 24.37 |
| 00000001 | Section Curio | 6 May 2022 | 11.63 |

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
