# Peer review of "Estimating Suspended Sediment Concentration Using Remote Sensing for the Teles Pires River, Brazil"

_sustainability, doi:10.3390/su15097049_

Round 1

Reviewer 1 Report

This study used remote sensing data to estimate suspended sediment concentration (SSC) with an empirical model. The manuscript looks like a report rather than a research article as many unnecessary details are described and presented. There are many studies that focus on SSC estimation from various remote sensing data, such as MODIS, VIIRS, and Landsat/Sentinel-2 etc. Compared to previous studies, the improvement and innovation is very limited in modeling method and spatio-temporal analysis. On the other hand, a small number of samples were used, indicating that the proposed empirical model may be unstable. In addition, Figure 5-6 show model fitting results, and validation results are missing. Independent validation is an important step for ensuring the accuracy of water quality estimation.

According to above reasons, this manuscript should be rejected for publication.

Author Response

This study used remote sensing data to estimate suspended sediment concentration (SSC) with an empirical model.

The manuscript looks like a report rather than a research article as many unnecessary details are described and presented.

Nowadays, the use of remote sensing to study the surface of the planet has become common, but when we want to evaluate water surfaces we must take into account some factors that are essential for the study of this type of surface, and less relevant for the study of the terrestrial surface. The work describes how this care was used in a balanced way, addressing the main issues of each stage.

The text has a writing style, but seeks to report with the proper technique and care of scientific research, with the perspective of demonstrating an applied science.

There are many studies that focus on SSC estimation from various remote sensing data, such as MODIS, VIIRS, and Landsat/Sentinel-2 etc. Compared to previous studies, the improvement and innovation is very limited in modeling method and spatio-temporal analysis.

Most of the works that propose the empirical estimation of SSC, through orbital remote sensing, are faced with the difficulty of obtaining images for the same day of the in situ collections, these difficulties increase as atmospheric obstructions occur and the size of the water bodies studied decreases. The applications of this type of investigation in water bodies, such as the Teles Pires River, are limited by the size of the sections where the SSC survey takes place, which vary from 60 to 160 meters, making it impossible to use orbital sensors such as the MODIS they have spatial resolution of 250 meters and temporal resolution of 1 to 2 days. To overcome such obstacles, orbital sensors such as the MSI, which has a spatial resolution of 10 meters, can be used, however its 5-day temporal resolution reduces the probability of images occurring on the same day as the in situ collections.

Most authors overcome these difficulties by using images from days after or before the field collections, and to validate the use of images, the authors observe flow variations, as if the flow undergoes small changes between the date of the image and the date of the field collection, the image is considered suitable for the study. However, there is a phenomenon that can cause divergences when using this approach, its name is hysteresis. The increase in SSC is linked to the increase in flow, but the increase in SSC is not immediate, as there is a time gap between the increase in flow and the increase in SSC, this delay is called hysteresis. Therefore, it is not possible to guarantee that the concentration of suspended sediments did not vary using only the flow rate as a reference. That is why the use of images from the same day of field collections is so important, even if this limits the number of samples.

Another advance of this work is linked to a clear methodology for extracting the values of the pixels, in which the methodology prevents contributions from the bottom and the margin, and describes the sedimentological behavior along the section through a single value, which is the average. This simplification of the behavior for the average occurs in a similar way in traditional measurements, where samples are collected at fixed intervals in the section, in which the samples are united, homogenized and submitted to the filtering process, and indirectly the methodology used in the work follows the same standard of representativeness of the traditional measures section.    

On the other hand, a small number of samples were used, indicating that the proposed empirical model may be unstable.

The small number of images was a choice to ensure that the reflectances in the images represented as real as possible the field collections. By only using images from the same day we can guarantee the quality of the sample. Most works report a number of images from days before or after the SSC field measurements, and therefore present a greater number of points in the relationship of the models, but our relatively unpublished work, despite the small number of samples, presented indices with satisfactory fit, and this is a proposal for science to explore and credit or not with other work.

In addition, Figure 5-6 show model fitting results, and validation results are missing. Independent validation is an important step for ensuring the accuracy of water quality estimation.

To solve this problem a figure comparing the observed data with the estimated data was elaborated for the validation data.

<See Figure 8 in revised draft>

According to above reasons, this manuscript should be rejected for publication.

We believe that the reviewer may reconsider his decision, considering our clarifications and our presented justifications.

Reviewer 2 Report

1.      The No. of images in figure 2 is inconsistent with table 2, figure 5, figure 6, and appendix A. The data in Figures 5 and 6 are selected from table 2 and appendix A for analysis, right? How to decide which data is valuable to study?

2.      Why some data in table 2 are the same?

3.      The collected SSC values are between 5 to 25, resulting in the statistical parameters showing relatively small values. So, it is hard to realize the accuracy. Therefore, I suggest adding one parameter of relative error % [=(estimate-measured)/ measured)] to reveal the error percentage.

4.      Please explain how to get the SSC data on the author’s part. Due to the SSC data varying with the time and location (longitudinal and cross-section), the SSC value is changing. However, the images in figure 2 are the date. So, how to connect the SSC value with satellite images simultaneously and at the exact location? The method in figure 3 seems to take an average of satellite images of the cross-section. However, the SSC values at the main channel differ from the bank area.

5.      In conclusion, the contents suggest stating the limitation and valuable analysis range of SSC values from satellite images.

Author Response

  1. The No. of images in figure 2 is inconsistent with table 2, figure 5, figure 6, and appendix A. The data in Figures 5 and 6 are selected from table 2 and appendix A for analysis, right? How to decide which data is valuable to study?

The authors investigated and corrected the inconsistencies.

In Figure 2 some images were missing, and they have been added. The data in Appendix I are all data from the hydrological stations that had solid discharge data, but only the field campaigns that took place on the same day as the satellite passes were used to create the models.

  1. Why some data in table 2 are the same?

Some SSC collection stations are located at the intersection between two tiles (of the images), which provided two images for the same day. When evaluating the spectral responses without atmospheric calibration, the values are different, but very close to each other. After calibration, the ACOLITE procedure further equalizes the spectral responses. By increasing the precision of the data in the table, the differences become more noticeable, according to the better presentation of the data (greater precision) in Table 2.

  1. The collected SSC values are between 5 to 25, resulting in the statistical parameters showing relatively small values. So, it is hard to realize the accuracy. Therefore, I suggest adding one parameter of relative error % [=(estimate-measured)/ measured)] to reveal the error percentage.

The parameter was added to the models evaluation process, through equation 6, and the results presented in Table 2, demonstrating what the reviewer suggested.

  1. Please explain how to get the SSC data on the author’s part. Due to the SSC data varying with the time and location (longitudinal and cross-section), the SSC value is changing. However, the images in figure 2 are the date. So, how to connect the SSC value with satellite images simultaneously and at the exact location? The method in figure 3 seems to take an average of satellite images of the cross-section. However, the SSC values at the main channel differ from the bank area.

Elements were added in the work that explain the SSC sampling methodology through EWI, demonstrated in the description of the EWI method and in Figure 2.

  1. In conclusion, the contents suggest stating the limitation and valuable analysis range of SSC values from satellite images.

Yes, this statement was discussed and also reported in the conclusions of the work, highlighting the range of SSC values found and validated by the models.

Reviewer 3 Report

Report on “Estimating Suspended Sediment Concentration using Remote Sensing for the Teles Pires River, Brazil”

Suspended sediment concentration for the Teles Pires River in Brazil’s Amazon is estimated by Remote sensing in this manuscript. Access to several data sources and processing robustness show that Google Earth Engine can be a good tool to accurately estimate water quality parameters via remote sensing.

This manuscript deals with an interesting and up-to-date topic. To publish this, consider the following comments:

1. The author should add more content to the discussion section to improve the quality of the manuscript.

2. It may be more intuitive to present the comparisons to previous research comparison in the form of figure.

3. I expect the authors to present quantitative results in the conclusion section of the manuscript to highlight the findings of this study.

4. Improve the quality of charts.

Author Response

This manuscript deals with an interesting and up-to-date topic. To publish this, consider the following comments:

  1. The author should add more content to the discussion section to improve the quality of the manuscript.

Figures were added, the resolutions of others were improved, details were added in the tables and the discussion was also improved.

  1. It may be more intuitive to present the comparisons to previous research comparison in the form of figure.

We were unable to make presentations with figures that correlated other works, but we added details in the figures, especially in Figure 9, where we applied the B4 exponential model to the set of images for validation.

  1. I expect the authors to present quantitative results in the conclusion section of the manuscript to highlight the findings of this study.

The indices of the determination coefficient (R2) and the Nash-Sutcliffe efficiency (NSE) index were added and discussed.

  1. Improve the quality of charts.

These have been improved.

Round 2

Reviewer 1 Report

At least, I suggest that more in situ measurements should be collected for developing a stable estimation model. Although authors give some explanations on my previous comments, this manuscript still lacks innovation and the developed model is unstable for pratical application due to small samples. Thus, this manuscript should be rejected for publication in sustainability.

Author Response

Comments and suggestions for authors:

At the very least, I suggest that more in situ measurements are collected to develop a stable estimation model. Although the authors give some explanations about my previous comments, this manuscript still lacks innovation and the developed model is unstable for practical application due to small samples. Thus, this manuscript must be rejected for publication on sustainability.

The objective really explained in this work is to look for models that relate characteristics of satellite images (reflectances and indices) to the water quality parameter, which is the suspended sediment concentration (SSC) in different sections along the Teles Pires River which is influenced by different characteristics of the watershed (e.g., soils, topography, land use, management practices, precipitation, etc.).

This in itself already implies that the conception of a model for this basin and the conditions present may be unstable, due to the interaction that may have occurred with the variables that were presented in the results, and to the innovative approach that we established where field collection is matched with images from the same day, which is not common in other works.

A clear example of the variability that can occur for the same river, but with different conditions (model for only one hydrosedimentometric section) were the results found by Simöes et al. 2021 studying two sections of the Teles Pires River, downstream and upstream of a dam. These researchers found statistical significance indices of R² = 0.66 for one Upstream station and R² = 0.70 for another Upstream station. Using the downstream dam and the efficiency of the data estimated by the NDWI index, the model was categorized as “satisfactory.” This categorization was different from ours, which had “moderate” results for the reflectances of the red (B4) and green (B3) band, with linear and exponential models. The conclusion of the work by Simöes et al. 2021, as well as ours, is neither definitive nor stable, but it is the search for geoprocessing tools that can replace or complement traditional collections, allowing for an evolution in science and support for continuous monitoring. This can change as more research is developed and published, so that future researchers can show the most relevant factors in these relationships.

Thus, we sought to add more data, requesting such data from the responsible agency in Brazil, the National Water Agency (ANA). This increased added one more observation with the established coincidence condition (SSC x image). We then reviewed the image treatments to check for errors. Finally, we made a better distribution between the calibration and validation data, in order to represent the entire watershed of the Teles Pires river, and thus seek possible stability with regard to the conditioning variables of SSC.

To address concerns brought up by reviewers, we sought to obtain more data from the ANA to better fit the model, and at the same time, we sought to revise the methodology that would enable the stability of the model, and then we made some methodological changes to the article original.

When we analyzed the hydro-sedimentological data from 2016 to 2020, from the entire Teles Pires River basin, and also the data we collected in the field in the Curio Section (section that we monitored in 2021 and 2022), we obtained 19 pairs of data, which had field collections (SSC) coinciding with the passage of the satellite at the sampling site (image of the collection section), which is the main objective of our work, in the construction of an innovative model.

When requesting more data (to match the number of SSC samples from the field) from the National Water Agency of Brazil (ANA), which is responsible for coordinating the hydro-sedimentological data for our research area, it gave us only the data for the year 2021 in this time interval, as they have not yet analyzed the consistency of the data collected in 2022.

Thus, with these new data, we were able to find 3 more pairs of data for the entire Teles Pires River basin, with SSC measurements and images from the same day.

In this new approach we produced a model with 22 images, but with the treatment of the images, we only got 1 more pair of data, obtaining a maximum of 20 pairs of data (SSC x images on the same day).

These inconsistencies are due to several factors, such as:

  • Images were reprocessed taking into account a broader reference area for atmospheric calibration and sunglint removal.
  • A new evaluation of the images was elaborated, segmenting the images that did not have obstruction by clouds. In this approach, we produced a model with 22 images. However the image from 01/11/2018 for the UHE Teles Pires station downstream was observed as an anomaly. The reflectance of the aquatic and non-aquatic targets presented about half the expected reflectance. Atmospheric correction was not efficient in minimizing atmospheric contributions to the image. For the same station, the image from 07/19/2019 had a sudden increase in flow, significantly changing the reflectance conditions.

Thus, despite initially having a new model with 22 images, after a long process of image treatment and investigation we discovered that we only had one more image left, resulting in 20 pairs of data, or points to establish the model.

With the addition of one more image, the data showed the same pattern of normality and correlation.

Please review the uploaded file titled “Paulista et al_2218825_Responses to Reviewers_2round_Final3.docx” showing how improved methods using same day images as field data have better fit compared to images taken up to 2 days later than field data.

Reviewer 2 Report

1.      Please add a more detailed description of “ V’ which is presented in figure 2. Is it the meaning of vertical volume or vertical SSC values? It seems the sampled volume.

However, the SSC in the different depths of water could be various. So, can we say it is a “depth average SSC”?

2.      Based on the analysis timing (the same day, but maybe the different hour or different minute duration) of satellite images and sampled data. I suggest giving an assumption in section 2.2 for the flow pattern as a quasi-steady flow during satellite images and sampled data on the same day.

Author Response

Comments and Suggestions for Authors:

  1. Please add a more detailed description of “V” which is presented in Figure 2. Is it the meaning of vertical volume or vertical SSC values? It seems the sampled volume.

The "V" shown in Figure 2 means the path of descent and ascent of the suspended sediment sampler, which, having to present the same transit speed, has different sampling times in each vertical, and therefore samples different volumes of SSC in each vertical, but that each volume of the vertical will compose the total volume of the section.

However, the SSC in the different depths of water could be various. So, can we say it is a “depth average SSC”

Yes. The suspended sediment concentration (SSC) is variable in the different verticals and also in the depth of the same vertical, and this equal increment width (EWI) sampling method results in a composite sample of the monitoring section.

  1. Based on the analysis timing (the same day, but maybe the different hour or different minute duration) of satellite images and sampled data. I suggest giving an assumption in section 2.2 for the flow pattern as a quasi-steady flow during satellite images and sampled data on the same day.

Yes. We complemented the text by better detailing that the SSC samples were taken in an average time interval of the satellite imagery time, and evaluating the non-occurrence of precipitation downstream of the section, during the collection time.